# Aerosol-enhanced high precipitation events near the Himalayan foothills

Goutam Choudhury[1,2], Bhishma Tyagi[1], Naresh Krishna Vissa[1], Jyotsna Singh[3], Chandan Sarangi[4], Sachchida Nand Tripathi[5], and Matthias Tesche[2]

[1]Department of Earth and Atmospheric Sciences, National Institute of Technology Rourkela, Rourkela 769008, Odisha, India
[2]Leipzig Institute for Meteorology (LIM), Leipzig University, Stephanstrasse 3, 04103 Leipzig, Germany
[3]Shanti Raj Bhawan, Paramhans Nagar, Kandwa, Varanasi 221106, India
[4]Department of Civil Engineering, Indian Institute of Technology Madras, Chennai 600024, India
[5]Department of Civil Engineering & Department of Earth Sciences, Indian Institute of Technology Kanpur, Kanpur 208016, India

**Correspondence:** Bhishma Tyagi (bhishmatyagi@gmail.com)

**Abstract.** Particulate emissions can alter the physical and dynamical properties of cloud systems and in turn amplify rainfall events over orographic regions downwind of highly polluted urban areas. The Indo-Gangetic Plains, one of the most polluted regions of the world, is located upwind of Himalayan foothills. The region, therefore, provides an opportunity for studying how aerosol effects in connection with orographic forcing affect extreme rainfall events. This study uses 17-years (2001–2017) of observed rain rate, aerosol optical depth (AOD), meteorological re-analysis fields, and outgoing longwave radiation to investigate high precipitation events at the foothills of the Himalayas. Composite analysis of all these collocated datasets for high precipitation events (daily rainfall > 95 percentile) is done to understand the inherent dynamics and linkages between AOD and extreme events. Clear and robust associations are found between high precipitation events, high aerosol loading and high moist static energy values. Results show an average increase in AOD by 36 % along with an average increase in low level moist static energy (1000–850 hPa) by $\approx 1500$ J Kg$^{-1}$ inside the domain for high precipitation events. The finding highlights the crucial role of the aerosol direct radiative effect on high precipitation events over the Himalayan region.

## 1 Introduction

Studies related to aerosols and their impact on weather and climate are crucial for understanding the anthropogenic effect on the Earth system (Stocker et al., 2013). Aerosols can affect the atmosphere directly by interacting with incoming and outgoing radiation, known as Aerosol-Radiation-Interactions (ARI) (Vinoj et al., 2014; Kant et al., 2017; Sarangi et al., 2018). Aerosols can modify cloud microphysical and macrophysical properties (Fan et al., 2016; Kant et al., 2019) by means of indirect effects, also known as Aerosol-Cloud Interactions (ACI) and ultimately influence precipitation and the water cycle.

Several studies have identified the role of aerosols in invigorating cloud systems and – under certain environmental conditions – resulting in extreme precipitation events (Tao et al., 2012; Yang and Li, 2014; Altaratz et al., 2014; Fan et al., 2016, 2018). The indirect effect of pollution aerosols on clouds formed over orographic regions has also been explored (Van den Heever and Cotton, 2007; Carrió et al., 2010; Han et al., 2012; Zubler et al., 2011; Fan et al., 2014, 2017; Xiao et al., 2015; Choudhury et

al., 2019). These studies suggest that aerosols may increase the magnitude of orographic precipitation by modifying the cloud microphysics (ACI). The modelling results of Fan et al. (2015) show that absorbing aerosols from the polluted Sichuan Basin by virtue of ARI can increase the magnitude of high precipitation over mountainous regions downwind through aerosol-enhanced conditional instability (AECI). This process can lead to disastrous floods. The AECI mechanism occurs in two steps:

First, the presence of absorbing aerosols in the urban polluted basin increases the lower level stability during the daytime, thereby limiting the moisture in the lower levels. As a result, the moist static energy (MSE) gets accumulated in the low levels. Second, during nighttime, this accumulated MSE is transferred towards the mountains downwind where the now more humid air mass undergoes orographic lifting which results in anomalously high precipitation over the mountains. This mechanism connects aerosol direct and indirect effects to high precipitation events over mountainous regions. This mechanism is illustrated in the Figure 5 of Fan et al. (2015).

The Indian subcontinent receives most of its rainfall during the monsoon season when the branch of Inter Tropical Convergence Zone (ITCZ) synonymously termed as the monsoon trough carries large amounts of moisture from the ocean inland (Krishnamurti, 1985; Kumar et al., 1995; Benn and Owen, 1998; Gadgil, 2003; Gadgil et al., 2005). An increasing trend of extreme precipitation events has been reported over the Himalayas (Dimri et al., 2017; Bohlinger and Sorteberg, 2018; Shrestha et al., 2019). The occurrence of these extreme rainfall events over the steeper mountains can result in catastrophic floods and landslides with devastating effects on the population. The majority of these rainfall events are embedded within large-scale synoptic weather systems such as monsoon depressions, low-pressure systems, and occasional extratropical disturbances (Dhar and Nandargi, 1993; Rajeevan et al., 2010; Nandargi and Dhar, 2011; Bohlinger et al., 2017; Dimri et al., 2017). These synoptic features result in a strong cross-orographic flow which favours the orographic lifting of moist air parcels.

The Indo-Gangetic Plain (IGP) lies to the south of the Himalayan Foothills and is associated with high aerosol loading (Ramachandran et al., 2012; Dahutia et al., 2018; Reddy et al., 2018), with a substantial contribution of absorbing aerosols like black carbon (Rana et al., 2019; Brooks et al., 2019) and dust (Lau et al., 2010; Gautam et al., 2013; Lau et al., 2017). A recent modelling study by Lau et al. (2017) has revealed the crucial role of ARI in modifying the monsoon circulation resulting in monsoon onset over the Himalayan Foothills by one to five days for the year 2008. They also found that the aerosols by semi-direct effect increase the stability of the lower troposphere in regions south of the Himalayas and weaken the convection (similar to the findings of Fan et al. 2015). This non-precipitated moisture when gets advected by the monsoon southerlies can increase the rainfall over the Himalayas. However, their study emphasized on the whole monsoon season and not on extreme precipitation events. Because of the computational limitations, modelling studies relating the aerosol and precipitation are restricted to case studies. However, with the availability of reanalysis datasets, e.g. MERRA-2 (Modern Era Retrospective analysis for Research and Applications–version 2) (Bosilovich et al., 2015) and ERA (European Centre for Medium-Range Weather Forecasts Re-Analysis) Interim (Dee et al., 2011) data products, statistical analysis over a long period is possible to unveil the relations between aerosols and meteorology. With the defined relationship between aerosols and MSE for extreme precipitation events (Fan et al., 2015) in Sichuan Basin, China, and availability of reanalysis fields and satellite derived aerosol optical depth (AOD) data, one unaddressed question is if the aerosols via ARI have a role in the high precipitation events over

the Himalayan region? The present study attempts to find this answer by analysing the AOD and meteorological fields for the high precipitation events occurring over the Himalayan foothills.

The paper is organised as follows: The study area description and data used for the study are described in Section 2, while Section 3 discusses the methodologies adopted in the present work. Section 4 presents the results obtained in the context of AECI and the discussion of its linkages with previous findings. Concluding remarks are given in Section 5.

## 2 Study domain and data used

The present study investigates the possible connection between aerosol load and high precipitation events over a selected Himalayan region for a period of 17 years from 2001 to 2017. The chosen domain shown in Figure 1, is close to the IGP and bounded by the coordinates (29 °N, 78.5 °E), (26.7 °N, 83 °E), (28.7 °N, 84 °E) and (31 °N, 79.5 °E), with average, maximum and minimum elevations of 1649 m, 5581 m and 84 m respectively. The selection of the domain is inspired by the work of Bohlinger et al. (2017) where they used 30 years of climatological rain-gauge data to form a cluster of stations with similar rainfall pattern.

For information on aerosol load, this study utilizes 1 ° x 1 ° daily gridded combined Dark Target Deep Blue (DT-DB) aerosol optical depth (AOD) at 550 nm as provided in the Moderate Resolution Imaging Spectro-radiometer (MODIS) level 3 collection 6.1 aerosol data product for Terra overpasses at about 10:30 am. This product combines AOD as inferred using the Dark Target (Levy et al., 2007, 2013) and Deep Blue (Hsu et al., 2013) algorithms to provide improved data coverage over both dark and bright land surfaces (Bilal et al., 2018; Wei et al., 2019). From here onwards, we will refer to MODIS AOD instead of combined DT-DB MODIS AOD. Very intense precipitation is accompanied by optically thick clouds which inhibit an AOD retrieval from MODIS observations. However, the extreme precipitation events in the study domain occurred mostly during nighttime while the considered Terra overpasses are fixed to approximately 10:30 am local time. This time difference enabled us to find cases with valid AODs. Table S1 in the supplementary material lists the considered dates of extreme events, the domain averaged daily precipitation, the domain averaged AOD derived from MODIS-TERRA combined dark target deep blue algorithm and MERRA-2 together with the percentage of grid points with valid AODs within the study domain for each event. Supplementary Figure S1 shows the percentage of data with valid AODs at every grid point used to compute the AOD composite as presented in this work.

Precipitation data is derived from Tropical Rainfall Measuring Mission version-6 (TRMM 3B42) gridded data product with a spatial resolution of 0.25° x 0.25° and a temporal resolution of 3 hours (Huffman et al., 2007). The rainfall rate derived from TRMM 3B42 data is comparable with surface observation in terms of spatial variations but not in terms of magnitudes (Koo et al., 2009; Sapiano and Arkin, 2009; Ochoa et al., 2014; Iqbal and Athar, 2018). The meteorological variables such as temperature, geopotential height, relative humidity and specific humidity, Boundary Layer Height (BLH), and Convective Available Potential Energy (CAPE) are derived from ERA Interim and MERRA-2 reanalysis data. The AOD from MERRA-2 is also used to study the aerosol load. National Oceanic and Atmospheric Administration (NOAA) daily Outgoing Longwave Radiation (OLR) data is used to identify the convection and cloud patterns over India (Liebmann and Smith, 1996). All data

sets used in the present study are considered for the monsoon season (June–September) for 17 years (2001–2017). Table 1
summarizes the data sets used in the present study along with the source and resolution.

## 3   Methodology

The domain-averaged daily accumulated precipitation is used to define the High Precipitation (HP) events as those days that
exceed the 95-percentile value of 32.84 mm. This criterion is used in many studies (Indrani and Al-Tabbaa, 2011; Varikoden
and Revadekar, 2019) to categorize the heavy rainfall events over the Indian subcontinent. The threshold value is similar to that
of Bookhagen (2014) but smaller than the measurements from in-situ rain gauge stations (Nandargi and Dhar, 2011; Bohlinger
and Sorteberg, 2018). The lower value may be due to the underestimation of the precipitation magnitude over areas of high
surface elevation by TRMM (Scheel et al., 2011; Iqbal and Athar, 2018). Since an extreme weather system may stay active
for more than one day, we further constrain the HP events such that a gap of at least two days exists between two consecutive
events. The total number of 75 events with mean and maximum precipitation of 45.05 mm and 90.2 mm, respectively, were
found from this analysis.

MSE is a measure related to moist convection (Kiranmayi and Maloney, 2011; Mayer and Haimberger, 2012; Fan et al.,
2015) and can be computed for each pressure level as:

$$MSE = C_\mathrm{p}T + Lq + Z \tag{1}$$

where $T$ is the temperature, $Z$ is the geopotential, $L = 2{,}260\,\mathrm{kJ\,kg^{-1}}$ is the latent heat of vaporisation, $C_\mathrm{p} \approx 1\,\mathrm{kJ\,kg^{-1}\,K^{-1}}$ is
the specific heat capacity at constant pressure and $q$ is the specific humidity.

The lower level MSE is calculated by averaging the MSE for pressure levels between 1000 hPa to 850 hPa at 00 UTC or
05:30 local time. The AOD, daily accumulated precipitation, OLR, boundary layer height (BLH) anomaly, convective available
potential energy (CAPE) anomaly and the MSE composites are evaluated for the selected high precipitation events and up to
two days prior to the event day. The composite of a parameter is calculated by averaging over all the selected high precipitation
events. The anomaly of a parameter is calculated by first computing the anomaly for each event day and then averaging over
all the events. The anomaly for an event day is calculated by subtracting the 17 year average value of the parameter from the
value of the parameter for that day.

## 4   Results and Discussion

The composite analysis is widely used to study the meteorological features of different systems (Krishnan et al., 2000; Rajee-
van et al., 2010; Pillai and Sahai, 2014; Bohlinger et al., 2017). This enables the identification of physical processes connected
to the spatial variation of different atmospheric parameters. Figure 2 depicts the composites of AOD (derived from MODIS
and MERRA-2) and low level MSE (derived from ERA-Interim and MERRA-2) for event days, one day prior and two days
prior to the events. It shows a gradual increase in lower-level MSE within and near the domain, reaching the maximum value
on the event day with an average increase of about 1500 J kg$^{-1}$. Coincidentally, AOD shows a similar pattern with an increase

of 36% (24%) from the previous day to event day using MODIS (MERRA-2) data. The AOD and MSE variations are showing similar patterns by MERRA-2 data as well; however, the magnitudes are smaller compare to MODIS and ERA-Interim data. The lower values reported by MERRA-2 may be due to missing emission sources in the aerosol model (Buchard et al., 2017). The MERRA-2 dataset can capture the dynamic changes in AOD variations (Shi et al., 2019) and for the present study also; the dynamic variations in AOD and MSE are captured by MERRA-2 as that of MODIS and ERA-Interim. This hints towards a connection between the accumulation of aerosols and the increase in lower-level MSE within and around the domain. There is a statistically significant positive correlation between the domain averaged AOD and MSE for both MERRA-2 (Pearson correlation coefficient $R = 0.2649$, $p = 0.0235$) and ERA-Interim ($R = 0.352$, $p = 0.0023$) datasets. The presence of absorbing aerosols within the BLH may have warmed the lower atmosphere increasing the low level stability and suppressing the convection, thus preventing the consumption of MSE. This may have resulted in accumulation of MSE in the lower levels as seen in Figure 2. We repeated this composite analysis for other parts of the Himalayan Foothills and also for days with no or low precipitation (with the threshold of 5 percentile) but were unable to find a feature as observed in Figure 2 for high precipitation events (shown in supplementary Figure S2).

The composites of BLH and CAPE anomalies at 12 UTC are shown in Figure 3. The BLH is seasonally and temporally variable over any place, and in general, the convective days have higher values of BLH (Yang et al., 2014). The higher AOD values increase the reflection and absorption of solar radiation and the amount of radiation reaching to surface decreases. This reduced radiation, in turn, reduces turbulence generation and hence the BLH. For the present study, a negative BLH anomaly is found within and close to the domain, especially on the event day (see Figure 3). The negative BLH anomaly might be a signature of the accumulated aerosols stabilizing the boundary layer by reducing the amount of solar radiation reaching the surface during day time (Wendisch et al., 2008; Li et al., 2011). The reduction in incoming solar radiation to surface reduces the sensible and latent heat fluxes and may further enhance the accumulation of aerosols and moisture (Takemura et al., 2005) in the lower levels, acting as a cyclic process. The variation is similar to the modelling results by Fan et al. (2015), showing a decrease in BLH due to the radiative effects of the polluted urban aerosols of the Sichuan basin, China. The MSE may then get accumulated in the lower levels during day time by the AECI mechanism because of suppressed convection.

The increase in AOD is found to influence the lifetime of severe convective clouds with the availability of higher CAPE values (Chakraborty et al., 2016). CAPE signifies the potential of the convective event with higher values representing possibilities of severe events. The presence of absorbing aerosols within the planetary boundary layer may result in an increase in temperature in the lower atmosphere. This can result in higher CAPE values above the convection condensation level (Wang et al., 2013; Sarangi et al., 2015; Li et al., 2017). Please note that several studies have highlighted that Aerosol Invigoration effect (Rosenfeld et al., 2008; Fan et al., 2018) is a relevant phenomenon over this region during monsoon (Sarangi et al., 2017, 2018). Aerosol-induced microphysical changes (like smaller cloud droplets) cause the droplets to move higher up in the atmosphere instead of falling down as rain. As such, the ratio of ice to liquid water content in the clouds increase in such invigorated clouds. A higher CAPE (induced due to the aerosol radiative effect) can further add to the cloud invigoration and lead to more ice formation, and thus higher rain rate eventually. Moreover, higher CAPE anomalies on the severe event days in the presence of higher AOD values are representing the possibility of a severe event as the convection grows for a longer

time with more accumulation of MSE, CAPE and AOD, going to a positive feedback cycle. The composites of CAPE anomaly supportively show a positive value in the domain starting from one day prior to the event (about 400% increase). The higher values of CAPE anomaly during evening hours (17:30 local time) are similar to the variation reported by Fan et al. (2015). The association of higher CAPE with increased AOD values results in a high precipitation event over the domain as observed.

On the other hand, OLR represents the thick cloud cover over any region and the OLR variation is typically used to identify the area with severe convective clouds (Inoue et al., 2008). The daily accumulated precipitation and OLR anomaly composites show much lower precipitation rates over central India south of the selected domain compared to that at the Himalayas on and before the HP events (Figure 4). However, this low precipitation rate in the vicinity of the domain may support the working of an AECI mechanism by reducing the scavenging of aerosols through wet deposition, which enables the initial accumulation of aerosols for initiating the AECI effect in the first place. The variation of three-hourly domain averaged precipitation for the selected events with time (Figure 5) shows low precipitation during the day time over the domain, with an anomalous increase during the night time after around 15 UTC (20:30 local time). This pattern of rainfall showing weaker (stronger) convection during the day (night) time over the domain further supports the theory of AECI (Fan et al., 2015).

The AOD averaged over the domain for monsoon months is found to be 0.574, while it is 0.8 for the high rainfall event cases. The aerosols generated from local and neighbouring sources (IGP) may get transported by the winds towards the domain during HP events, accumulating the aerosols blocked by the high elevation mountains (Lau et al., 2017). Bohlinger et al. (2017) identified that 75% of the moisture during an HP event is transported from the land sources with an increase in moisture supply from the IGP. This increase in moisture supply maybe because of the AECI effect, which prevents the consumption of moisture by suppressing the convection during the day time. When accumulated moisture gets transported towards the Himalayas during the night, it may undergo orographic lifting and may provide additional moisture and aerosols to already formed convective cells and result in HP events.

The presence of high aerosol load (AOD $\approx 0.8$) in a profoundly moist environment may also contribute to the excitation of HP events in orographic regions by aerosol indirect effect (Fan et al., 2014, 2017; Xiao et al., 2015). Some studies suggest that under high AOD values with high black carbon concentration, the ARI effect dominates the ACI effect (Fan et al., 2015; Rosenfeld et al., 2008; Ding et al., 2019). Several studies exploring the indirect impact of anthropogenic aerosols on mixed-phase orographic clouds also found a minor impact on the net orographic precipitation (Saleeby et al., 2013; Fan et al., 2014; Letcher and Cotton, 2014). Negligible impact of aerosol indirect effect on the orographic precipitation over the foothills of Nepal Himalayas was also noticed by Shrestha et al. (2017). However, the exact contribution of the direct and indirect effect of aerosols to the HP events is not within the scope of the present study and requires further work involving state of the art numerical models.

## 5 Conclusions

To summarise, we selected HP events (defined based on domain averaged daily accumulated precipitation) over the Himalayan foothills during monsoon months for a period of 17 years to investigate a possible connection with aerosols. The composites of

AOD, lower-level MSE, BLH, and CAPE anomaly, daily accumulated precipitation and OLR anomaly are analyzed to identify the dynamical progression of these parameters impacting HP events. The findings can be concluded as follows:

1. The AOD within the domain for the event days increased by about 36% compared to that of the previous day.

2. Spatio-temporal pattern of low-level MSE coincides with that of the AOD for the selected events. It may be a result of the warming effect of aerosols in the lower atmosphere which increases (decreases) the stability in the lower (mid) levels. This mechanism is further supported by the occurrence of negative BLH anomaly and high positive CAPE values within the selected domain. This indicates that the aerosols can have a substantial impact in increasing the magnitude of the orographic precipitation by AECI effect.

3. The variation of 3 hourly rainfall for the event day shows that the HP event over the domain happens mostly during the night.

4. The spatial distribution of daily accumulated precipitation composite shows low values close to the domain for one and two days prior to HP event. Thus, it may support the initial accumulation of aerosols before HP events within and in the vicinity (IGP) of the domain and maybe an essential requirement for the working of AECI mechanism.

The aerosols accumulate near and within the selected domain starting from two days prior to the event. The low precipitation as seen in Figure 4 before the events may have allowed this build-up of aerosols. By the AECI mechanism, the convection in the lower atmosphere is suppressed during the day time preventing the consumption of the moist static energy. This excess energy, then gets transported towards the orography during night and leads to an increase in the magnitude of high precipitation events. We have schematically summarized the AECI mechanism for the Himalayan region in Supplementary Figure S3. The results show observational evidence of the aerosol enhanced convective instability mechanism identified by Fan et al. (2015). However, the analysis incorporates uncertainty due to the humidity growth effect, where the higher values of AOD might be a result of the hygroscopic growth of aerosols and not because of the presence of more aerosol particles. This is why we incorporated AOD values using MERRA-2 data. In the presence of thick cloud cover during the high precipitation event, the MODIS data will not be available. Thus, we did not have the AOD observations from MODIS at every grid point inside our domain for the individual events (supplementary Figure S1). The composite images have better data availability as visible in figures. To be certain, we cross-checked the pattern of variation for every case by MERRA-2 AOD. The results indicate that aerosols can play a vital role in exciting HP events over the Himalayas in monsoon season. Thus, aerosols including the chemistry are essential to consider for forecasting HP events over the Himalayan region in the regional modelling studies.

*Data availability.* All datasets used in this work are open source and publicly available. The MODIS L3 daily gridded aerosol data is available at the NASA Goddard Space Flight Center (GSFC) and Atmosphere Archive and Distribution System (LAADS) (https://ladsweb.modaps. eosdis.nasa.gov/). The TRMM precipitation data is available from NASA/Goddard Space Flight Center and PPS, archived at the NASA Goddard Earth Sciences (GES) Data and Information Services Center (DISC)(https://disc.gsfc.nasa.gov/). The MERRA-2 reanalysis data

can be accessed from NASA GES DISC as well. The ERA-Interim reanalysis data can be accessed from ECMWF (https://www.ecmwf.int/) under forecast datasets. The Interpolated OLR data is provided by the NOAA/OAR/ESRL PSL, Boulder, Colorado, USA, from their Web site at https://psl.noaa.gov/.

*Author contributions.* GC, and BT conceived the study. GC, BT, NKV did data analysis, made the plots and wrote the initial manuscript. JS, CS, SNT, and MT improved the idea from time to time and corrected the manuscript versions. All authors have contributed to the research and took part in finalizing the manuscript.

*Competing interests.* The authors declare no competing interests.

*Acknowledgements.* Authors want to acknowledge Department of Science and Technology, Govt. of India for providing the funding (project-funding code: DST/CCP/Aerosol/90/2017(G), under SPLICE, Climate change, National Network Programme on Aerosol). The authors are grateful to NASA Goddard Space Flight Center (GSFC) and Atmosphere Archive and Distribution System (LAADS) for making the level-3 MODIS datasets available. Authors are thankful to ECMWF for providing ERA reanalysis datasets, and NASA Goddard Earth Sciences (GES) Data and Information Services Center (DISC) for MERRA-2 reanalysis data. The authors acknowledge the NASA/Goddard Space Flight Center and PPS, which develop and compute the TRMM precipitation data archived at the NASA GES DISC, as a contribution to TRMM project. The Interpolated OLR data is provided by the NOAA/OAR/ESRL PSL, Boulder, Colorado, USA and is highly acknowledged.

*Financial support.* GC and MT received funding from the Franco-German Fellowship Programme on Climate, Energy, and Earth System Research (Make Our Planet Great Again – German Research Initiative, MOPGA-GRI) of the German Academic Exchange Service (DAAD), funded by the German Ministry of Education and Research.

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

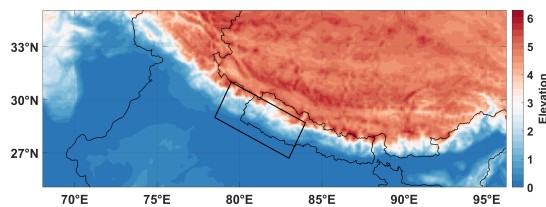

**Figure 1.** Map of Global Multi-resolution Terrain Elevation Data 2010 (GMTED2010) in kilometres provided by the US Geological Survey. Thin black lines mark country borders. The bold black line marks the domain at the foothills of the Himalayas used in this study.

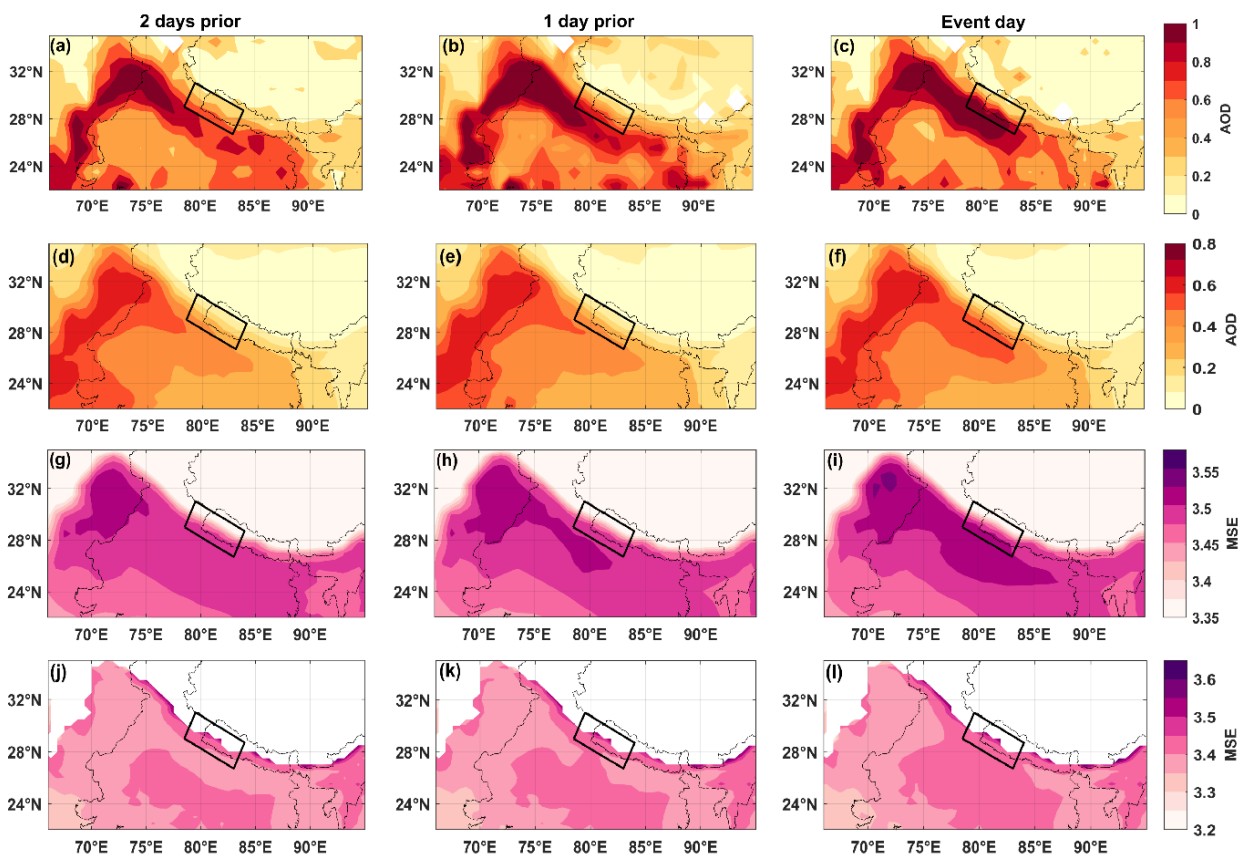

**Figure 2.** Composites of MODIS AOD (a-c), MERRA-2 AOD (d-f), ERA-Interim lower level MSE (g-i) in units of $10^5$ J kg$^{-1}$ and MERRA-2 lower level MSE (j-l) in units of $10^5$ J kg$^{-1}$, for the event days (c, f, i, l), one day prior (b, e, h, k) and two days prior (a, d, g, j) to the events.

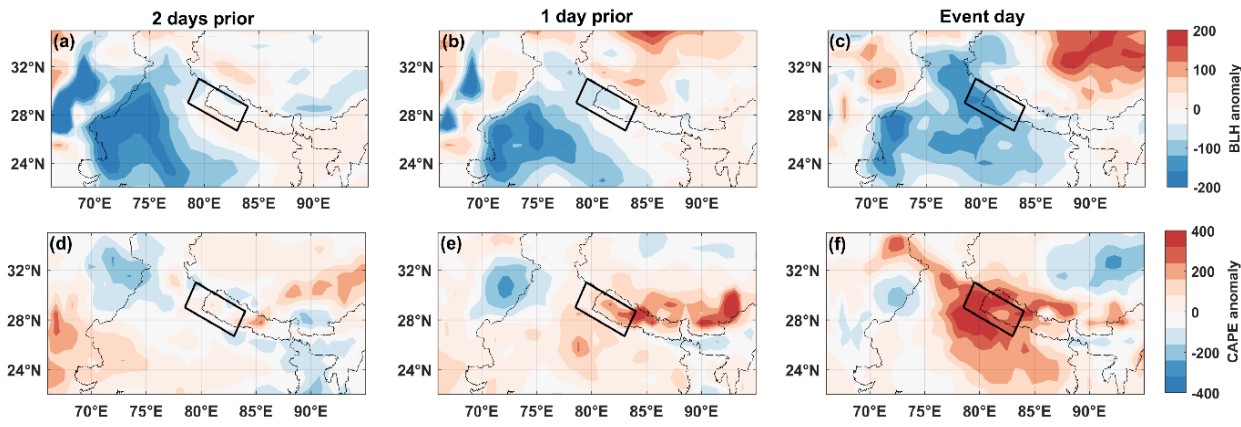

**Figure 3.** Composites of BLH anomaly (a-c) in units of meter for 12 UTC and CAPE anomaly in units of J kg$^{-1}$ (d-f) at 12 UTC, for the event days (c, f), one day prior (b, e) and two days prior (a, d) to the events.

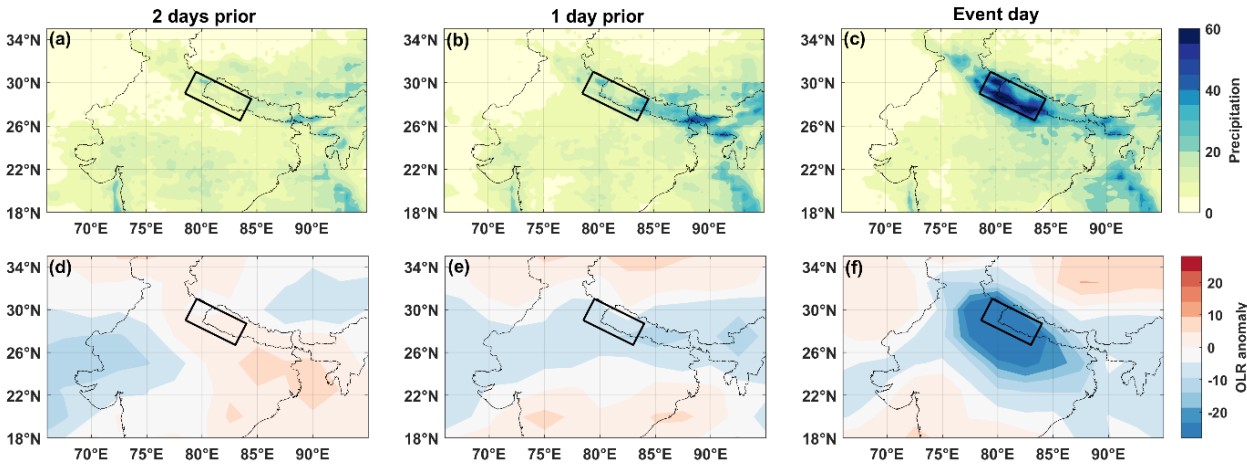

**Figure 4.** Composites of daily accumulated precipitation (a-c) in mm day$^{-1}$ and OLR (d-f) in W m$^{-2}$ for the event day (c and f), one day prior to the event (b and e) and two days prior to the event (a and d).

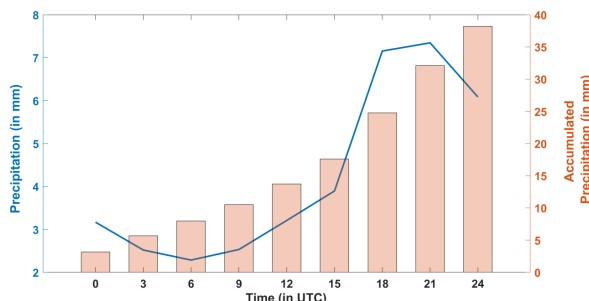

**Figure 5.** Domain-averaged three hourly precipitation on the day of occurrence of a high precipitation event averaged for all events. The accumulated precipitation values are given in background as bars.

**Table 1.** Overview of the data sets used in this study.

| Data Product | Data Source | Parameters | Resolution |
|---|---|---|---|
| MODIS Level-3 Collection 6.1 (MOD08 D3) | LAADS DAAC, NASA | AOD | Daily, $1° \times 1°$ |
| TRMM 3B42 V7 3-hourly precipitation | PMM, NASA | Precipitation rate | 3 hourly, $0.25° \times 0.25°$ |
| ERA INTERIM reanalysis | ECMWF | Geopotential, Humidity, BLH, CAPE | 6 hourly, $0.75° \times 0.75°$ |
| NOAA Interpolated OLR | NCEP, NOAA | OLR | Daily, $2.5° \times 2.5°$ |
| MERRA-2 AOD and meteorological fields | GES DISC, NASA | AOD, Humidity, Temperature, Pressure, Surface Geopotential Height | 3 hourly, $0.625° \times 0.5°$ |