# Peer review of "Aerosol-enhanced high precipitation events near the Himalayan foothills"

_Atmospheric Chemistry and Physics, 2020_

## Referee Comment (RC1) · Anonymous Referee #1 · 7 Jul 2020

To the Authors:

The data presented in the paper is precious and work is valuable for other researchers, however, the paper need to be thorough revision by keeping its value for science as well as language. I suggest following points to improve for the revised version of this work:

1. In line 60, you mention "aerosol concentration". Do you mean "aerosol optical depth"? If yes then please change it accordingly as they both have different meaning.

2. In the second section, you say that the domain selection is based on the work of Bohlinger et al. 2017. However, they have classified three domains in their study. Is there any specific reason of considering only one part of Himalayan foothills? Does the

[Figure]

AECI mechanism only works in the selected domain and not in other parts? Please explain this in the conclusion section to be clearer.

3. Typo in line 70: change "toMODIS" to "to MODIS".

4. The information regarding the successful AOD retrieval described in the second section from line 71 to 78 would make more sense in the result section after you describe the AOD composite plot.

5. In Figure-2, why are the color-bar limits different in the plots of AOD and MSE? Please make it same for better comparison.

6. Lines 123 & 124: Does "R" means Pearson's correlation coefficient? The methodology used in calculating correlation coefficient should be mentioned in the draft.

7. Typo in line 131: it should be BLH instead of PBLH.

8. The third paragraph of result section describes the association of aerosols and CAPE for extreme events. The effect is described; however the causation needs to be elaborated. Following papers can be referred and included in the manuscript.

Wang, Y., Khalizov, A., Levy, M. and Zhang, R., 2013. New Directions: Light absorbing aerosols and their atmospheric impacts. Atmospheric Environment, (81), pp.713-715.

Li, Z., Rosenfeld, D. and Fan, J., 2017. Aerosols and their impact on radiation, clouds, precipitation, and severe weather events. In Oxford Research Encyclopedia of Environmental Science.

9. The abstract needs modification. Could you please add some quantitative findings (AOD increase and MSE increase percentages) of this study in your abstract?

10. A schematic diagram showing the AECI mechanism over the Himalayan Foothill (similar to Fan et al., 2015) would better convey the message and ease the understanding of literature.

---

## Referee Comment (RC2) · Anonymous Referee #2 · 14 Jul 2020

The work is novel and the results are interesting. I feel that the manuscript should consider the following points:

Major Comments:

1) L 21- you mention "net orographic precipitation". Is it any different from "orographic precipitation"? Is there something that's missing?

2) Is there any specific reason for selecting the MSE values only for lower levels? The proper reason should be described in the manuscript. Why not mid or upper levels? Do they also show similar pattern of variation as lower level MSE?

3) L 121-122- you explain that there is a hint towards a connection between the aerosol accumulation and increase in lower level MSE within and around the domain because

of AECI effect. A comment on the cause of this connection after this point will help in better understanding.

4) First you show that AOD and MSE have similar pattern of variations and have statistically significant correlation. Then you try to verify it meteorologically by analyzing BLH, CAPE, OLR, and precipitation pattern. Why do you bring CAPE into picture? Is CAPE related to the aerosols in such scenarios? Is there any feedback process involved? Please include proper explanation in the manuscript.

5) The main inference from this study is that the aerosols enhance the "instability" and moisture supply providing the extra energy for extreme precipitation events over the Himalayan Foothills, downwind of polluted IGP. However, the title of this manuscript is "Aerosol-induced high precipitation events near the Himalayan foothills". The word "induced" can be misleading. Please change it to "enhanced" or something similar.

6) The conclusion is missing an overall explanation of AECI mechanism in context of the high precipitation events over the Himalayan region. The second point (L 182-184) describes it in very short, but it needs to be explained properly.

7) L 195- Please include a reference of Figure S1 of the supplementary.

Minor Comments:

1) The work by Fan et al. (2015) is the base of your present work on the Himalayan Foothills. However, the results from Fan et al. (2015) were based on modelling studies and it should be mentioned in the introduction when you discuss about AECI mechanism in L22.

2) L 57- "Section 3 discuss" to "Section 3 discusses"

3) L 63- the domain is "inspired from" to "inspired by"

4) L 70- "toMODIS" to "to MODIS"

5) L 94- "underestimation" to "the underestimation"

[Figure]

6) L 129- "general" to "general,"

7) L 138- "radiative" to "the radiative"

8) L 158- "ample" to "an ample" & "HP event" to "the HP event"

9) L 166- "cell" to "cells"

10) L 167- "large amount" to "huge amount"

11) L 189- "vicinity of the domain (IGP)" to "vicinity (IGP) of the domain"
* * *

---

## Referee Comment (RC3) · Kuvar Satya Singh (Referee) · 27 Jul 2020

1. Have you checked the 3 hourly precipitation after the event day? May be it continues increasing in the next days, which can be a reason of low extreme precipitation threshold. 2. The value of constants (Cp and L) used in equation-1 should be mentioned in the literature. 3. In line 108, you mention "anomaly of a parameter is calculated by first computing the anomaly for each event day and then averaging over all the events". Please also include an explanation of how you calculated the anomaly. 4. In lines 131-133, while discussing the boundary layer height anomaly, you describe the impact of high humidity values, during high AOD conditions, on the convection growth. This doesn't fit in the paragraph. In lines 191-192, you also explain the possibility of higher AOD due to high relative humidity values. How would you justify the two statements?

[Figure]

5. In line 157, what do you mean by "the available air parcel"? Does it mean the air parcel which accumulates moisture and aerosols during day time via AECI mechanism? Please modify it accordingly. 6. In line 194-195, you mention "we did not have the AOD observations from MODIS at every grid point inside our domain". What I understand is you did not have the MODIS AOD observations at every grid point inside your domain for the individual events and not after averaging or making composite. Please rephrase the sentence. 7. The conclusion is missing an overall explanation of the AECI mechanism in the context of the Himalayan region. After concluding the findings in four bullet points, please include an overall explanation of how these findings combinedly result in an extreme precipitating event over the Himalayan Foothills. 8. The site map depicting the topography of the region can be improved further with higher resolution one. Please consider the higher resolution one in the revised version

---

## Author Comment (AC1) · 31 Aug 2020

1. In line 60, you mention "aerosol concentration". Do you mean "aerosol optical depth"? If yes then please change it accordingly as they both have different meaning.

Reply: We thank the reviewer for pointing this out. We actually meant aerosol load. We have changed it to "aerosol load".

2. In the second section, you say that the domain selection is based on the work of Bohlinger et al. 2017. However, they have classified three domains in their study. Is there any specific reason of considering only one part of Himalayan foothills? Does the AECI mechanism only works in the selected domain and not in other parts? Please explain this in the conclusion section to be clearer.

[Figure]

Reply: We appreciate the reviewer's point. We have tested this hypothesis for the whole region; however, it worked more closely for the study domain. The limited AOD data may be a valid reason for this over other areas, and also the flow of winds transporting MSE and AOD to the regions differently. However, this needs further analysis of long term data before reaching any conclusions, and not in the preview of the present work. We now have included the following in the Results Section from line 129 to 131: "We repeated this composite analysis for other parts of the Himalayan Foothills and also for days with no or low precipitation (with the threshold of 5 percentile) but were unable to find a feature as observed in Figure 2 for high precipitation events (not shown)."

3. Typo in line 70: change "toMODIS" to "to MODIS".

Reply: We apologize for the typo. We have corrected it in the revised manuscript.

4. The information regarding the successful AOD retrieval described in the second section from line 71 to 78 would make more sense in the result section after you describe the AOD composite plot.

Reply: We appreciate the recommendation of the reviewer. However, we think it is better to discuss about the data quality used to make the composite images in the data section. We have also discussed about it in the conclusion section where we list the uncertainties of the study.

5. In Figure-2, why are the color-bar limits different in the plots of AOD and MSE? Please make it same for better comparison.

Reply: The AOD data is used from two sources (MODIS and MERRA-2) which use different methodologies to estimate the AOD. This leads to difference in the maximum AOD and, hence, in the set upper limit of the colorbar. The same is true for MSE which is derived from ERA-Interim and MERRA-2. We have tried to address this in the result section from line 119 to 124. We try to emphasize on the variations rather than magnitudes which can be clearly seen in the figures.

6. Lines 123 & 124: Does "R" means Pearson's correlation coefficient? The methodology used in calculating correlation coefficient should be mentioned in the draft.

Reply: Thank you for pointing this out. We used Pearson's correlation coefficient. We have modified the text accordingly.

7. Typo in line 131: it should be BLH instead of PBLH.

Reply: We apologize for the typo. We have corrected it in the revised manuscript.

8. The third paragraph of result section describes the association of aerosols and CAPE for extreme events. The effect is described; however the causation needs to be elaborated. Following papers can be referred and included in the manuscript. Wang, Y., Khalizov, A., Levy, M. and Zhang, R., 2013. New Directions: Light absorbing aerosols and their atmospheric impacts. Atmospheric Environment, (81), pp.713-715. Li, Z., Rosenfeld, D. and Fan, J., 2017. Aerosols and their impact on radiation, clouds, precipitation, and severe weather events. In Oxford Research Encyclopedia of Environmental Science.

Reply: We apologize for the confusion. We have included the two references and an additional reference (shown below). Sarangi, C., Tripathi, S.N., Tripathi, S. and Barth, M.C., 2015. Aerosol–cloud associations over Gangetic Basin during a typical monsoon depression event using WRF–Chem simulation. Journal of Geophysical Research: Atmospheres, 120(20), pp.10-974.

We have explained it in more details by adding the following in the manuscript from line 147 to 156. "The presence of absorbing aerosols within the planetary boundary layer may result in an increase in temperature in the lower atmosphere. This can result in higher CAPE values above the convection condensation level (Wang et at., 2013; Sarangi et al., 2015; Li et al., 2017). Please note that many recent studies have highlighted that Aerosol Invigoration effect (Rosenfeld et al., 2008; Fan et al., 2018) is a relevant phenomenon over this region during monsoons (Sarangi et al., 2017;

2018). Aerosol-induced microphysical changes (like smaller cloud droplets) cause the droplets to move higher up in the atmosphere instead of falling down as rain. As such, the ratio of ice to liquid water content in the clouds increase in such invigorated clouds. A higher CAPE (induced due to the aerosol radiative effect) can further add to the cloud invigoration and lead to more ice formation, and thus higher rain rate eventually. Moreover, higher CAPE anomalies on the severe event days in the presence of higher AOD values are representing the possibility of a severe event as the convection grows for a longer time with more accumulation of MSE, CAPE and AOD, going to a positive feedback cycle."

9. The abstract needs modification. Could you please add some quantitative findings (AOD increase and MSE increase percentages) of this study in your abstract?

Reply: We agree with the reviewer. We have added the following sentence in the abstract discussing about the quantitative increase in AOD and MSE. "Results show an increase in AOD by ∼36 % along with an increase in moist static energy by 1500 J/kg per total air mass for high precipitation events.

10. A schematic diagram showing the AECI mechanism over the Himalayan Foothill (similar to Fan et al., 2015) would better convey the message and ease the understanding of literature.

Reply: We are thankful to the reviewer for this suggestion. Though we believe that the AECI process is working as explained by Fan et al. (2015) and the diagram given by them is apt, we have added a simple schematic diagram in the supplement of our manuscript. The following text has been added to the manuscript in the line 206. "We have schematically summarized the AECI mechanism for the Himalayan region in Figure S3 in the supplementary file."

Please also note the supplement to this comment:
https://acp.copernicus.org/preprints/acp-2020-440/acp-2020-440-AC1-supplement.pdf

[Figure]

[Figure]

**Supplement:**

[revised manuscript text omitted]

*Figure S3. Schematic diagram representing aerosol-enhanced conditional instability over the Himalayan region (after Fan et al., 2015). Top panel represents the clean/low aerosols in the atmosphere with relatively lower moist static energy (MSE) values and bottom panel shows higher amounts of aerosols and accumulated MSE values. In the polluted case, high amounts of MSE build up over the foothills of the Himalaya during daytime due to suppressed convection compared to the clean atmosphere (top panel). The excess MSE in the polluted case is transported towards the Himalayan region by the winds, generating much stronger convection and precipitation over the foothills of the mountain at night. Acronyms: MSE (moist static energy), SW (shortwave radiation), SH (surface sensible heat flux), and LH (surface latent heat flux). The picture of the Himalaya in the right hand bottom corner is showing Gomukh peaks of the mountains (adopted from http://gbpihedenvis.nic.in/Glimpses_Himalaya_Photo_gallary.html).*

---

## Author Comment (AC2) · 31 Aug 2020

1. L 21- you mention "net orographic precipitation". Is it any different from "orographic precipitation"? Is there something that's missing?

Reply: We apologize for the confusion. By net orographic precipitation we mean the total precipitation over the orography (leeward side + windward side). We have removed the word "net" in the revised version.

2. Is there any specific reason for selecting the MSE values only for lower levels? The proper reason should be described in the manuscript. Why not mid or upper levels? Do they also show similar pattern of variation as lower level MSE?

Reply: As suggested by Fan et al. (2015), the moist static energy gets accumulated

in the lower levels following AECI mechanism. That is why we have shown the low level MSE composites. Nevertheless, we have also checked for mid and upper level MSE composites but could not find such pattern. We have modified the lines 26 - 29 in the Introduction Section as follows. "First, the suppression of convection in the urban polluted basin by absorbing aerosols during daytime reduces the consumption of low-level Moist Static Energy (MSE). Second, during nighttime, this accumulated low-level MSE is transferred towards the mountains downwind where the now more humid air mass undergoes orographic lifting which results in anomalously high precipitation over the mountains."

3. L 121-122- you explain that there is a hint towards a connection between the aerosol accumulation and increase in lower level MSE within and around the domain because of AECI effect. A comment on the cause of this connection after this point will help in better understanding.

Reply: We agree with the reviewer. We have modified the lines 126 - 129 of the manuscript as follows. "The presence of absorbing aerosols within the BLH may have warmed the lower atmosphere increasing the low level stability and suppressing the convection during the day time, thus preventing the consumption of MSE. This may have resulted in accumulation of MSE in the lower levels as seen in Figure 2."

4. First you show that AOD and MSE have similar pattern of variations and have statistically significant correlation. Then you try to verify it meteorologically by analyzing BLH, CAPE, OLR, and precipitation pattern. Why do you bring CAPE into picture? Is CAPE related to the aerosols in such scenarios? Is there any feedback process involved? Please include proper explanation in the manuscript.

Reply: We apologize for the confusion here. The absorbing aerosol present within the boundary layer may result in warming the low atmosphere and thus create a temperature difference between low and mid atmosphere – an increase in CAPE. We have added the following explanation along with three references in the manuscript in lines

"The presence of absorbing aerosols within the planetary boundary layer may result in an increase in temperature in the lower atmosphere. This can result in higher CAPE values above the convection condensation level (Wang et al., 2013; Sarangi et al., 2015; Li et al., 2017). Please note that many recent studies have highlighted that Aerosol Invigoration effect (Rosenfeld et al., 2008; Fan et al., 2018) is a relevant phenomenon over this region during monsoons (Sarangi et al., 2017, 2018). Aerosol-induced microphysical changes (like smaller cloud droplets) cause the droplets to move higher up in the atmosphere instead of falling down as rain. As such, the ratio of ice to liquid water content in the clouds increase in such invigorated clouds. A higher CAPE (induced due to the aerosol radiative effect) can further add to the cloud invigoration and lead to more ice formation, and thus higher rain rate eventually. Moreover, higher CAPE anomalies on the severe event days in the presence of higher AOD values are representing the possibility of a severe event as the convection grows for a longer time with more accumulation of MSE, CAPE and AOD, going to a positive feedback cycle."

5. The main inference from this study is that the aerosols enhance the "instability" and moisture supply providing the extra energy for extreme precipitation events over the Himalayan Foothills, downwind of polluted IGP. However, the title of this manuscript is "Aerosol-induced high precipitation events near the Himalayan foothills". The word "induced" can be misleading. Please change it to "enhanced" or something similar.

Reply: We agree with the reviewer on this and have changed the title to "Aerosol-enhanced high precipitation events near the Himalayan foothills"

6. The conclusion is missing an overall explanation of AECI mechanism in context of the high precipitation events over the Himalayan region. The second point (L 182-184) describes it in very short, but it needs to be explained properly.

Reply: We agree with the reviewer. We have elaborated the second point in the conclusion of the revised manuscript as follows. "Spatio-temporal pattern of low-level MSE

coincides with that of the AOD for the selected events. It may be a result of the warming effect of aerosols in the lower atmosphere which increases (decreases) the stability in the lower (mid) levels. This mechanism is further supported by the occurrence of negative BLH anomaly and high positive CAPE values within the selected domain. This indicates that the aerosols can have a substantial impact in increasing the magnitude of the orographic precipitation by AECI effect."

7. L 195- Please include a reference of Figure S1 of the supplementary.

Reply: We thank the reviewer for pointing this out. We have included the reference in the revised manuscript.

Minor Comments: 1. The work by Fan et al. (2015) is the base of your present work on the Himalayan Foothills. However, the results from Fan et al. (2015) were based on modelling studies and it should be mentioned in the introduction when you discuss about AECI mechanism in L22.

Reply: We thank the reviewer for pointing this out. We have modified this in the introduction Section in the line 23 as follows. "The modelling results of Fan et al. (2015} show that absorbing aerosols from the polluted Sichuan Basin by virtue of ARI can increase the magnitude of high precipitation over mountainous regions downwind through aerosol-enhanced conditional instability (AECI)."

2. L 57- "Section 3 discuss" to "Section 3 discusses"

3. L 63- the domain is "inspired from" to "inspired by"

4. L 70- "toMODIS" to "to MODIS"

5. L 94- "underestimation" to "the underestimation"

6. L 129- "general" to "general,"

7. L 138- "radiative" to "the radiative"

[Figure]

8. L 158- "ample" to "an ample" & "HP event" to "the HP event"

9. L 166- "cell" to "cells"

10. L 167- "large amount" to "huge amount"

11. L 189- "vicinity of the domain (IGP)" to "vicinity (IGP) of the domain"

Reply to 2-11: We are very thankful to the reviewer for pointing these minor grammatical errors in the manuscript. We have modified all of them in the revised one.

---

## Author Comment (AC3) · 31 Aug 2020

1. Have you checked the 3 hourly precipitation after the event day? May be it continues increasing in the next days, which can be a reason of low extreme precipitation threshold.

Reply: We appreciate the suggestion of the reviewer. We initially thought the same and checked the 3 hourly precipitation after the event day. We found that it gradually decreased to zero in the next few hours.

2. The value of constants (Cp and L) used in equation-1 should be mentioned in the literature.

Reply: We thank the reviewer for pointing this out. We have mentioned the values in

the revised manuscript in line 103.

3. In line 108, you mention "anomaly of a parameter is calculated by first computing the anomaly for each event day and then averaging over all the events". Please also include an explanation of how you calculated the anomaly.

Reply: We agree with the reviewer. We have added the following sentence in the manuscript from lines 110 and 111. "The anomaly for an event day is calculated by subtracting the 17 year average value of the parameter from the value of the parameter for that day."

4. In lines 131-133, while discussing the boundary layer height anomaly, you describe the impact of high humidity values, during high AOD conditions, on the convection growth. This doesn't fit in the paragraph. In lines 191-192, you also explain the possibility of higher AOD due to high relative humidity values. How would you justify the two statements?

Reply: We believe that the hygroscopic growth effect of relative humidity on the AOD values is persistent in highly humid situations like an extreme event. But we cannot disregard the humidity/moisture content as it is one of the most important fuel to every precipitating system. Since the higher MSE values also account for the higher moisture content (humidity - 'q' in eq. 1), we have accepted the reviewer's suggestion and removed the following statement in the revised manuscript. "The humidity values are found to be higher during the periods of higher AOD, which support the availability of more latent heat for the convection growth (Lou et al., 2019)."

5. In line 157, what do you mean by "the available air parcel"? Does it mean the air parcel which accumulates moisture and aerosols during day time via AECI mechanism? Please modify it accordingly.

Reply: We thank the reviewer for pointing this out. We have omitted the line 157 in the revised manuscript.

6. In line 194-195, you mention "we did not have the AOD observations from MODIS at every grid point inside our domain". What I understand is you did not have the MODIS AOD observations at every grid point inside your domain for the individual events and not after averaging or making composite. Please rephrase the sentence.

Reply: We thank the reviewer for this suggestion. We have modified the sentence as follows. "Thus, we did not have the AOD observations from MODIS at every grid point inside our domain for the individual events (supplementary Figure S1)."

7. The conclusion is missing an overall explanation of the AECI mechanism in the context of the Himalayan region. After concluding the findings in four bullet points, please include an overall explanation of how these findings combinedly result in an extreme precipitating event over the Himalayan Foothills.

Reply: We agree with the reviewer. We have included the following in the manuscript from line 202 to 206. "The aerosols accumulate near and within the selected domain starting from two days prior to the event. The low precipitation as seen in Figure 4 before the events may have allowed this build-up of aerosols. By the AECI mechanism, the convection in the lower atmosphere is suppressed during the day time preventing the consumption of the moist static energy. This excess energy, when gets transported towards the orography during night leads to an increase in the magnitude of high precipitation events."

8. The site map depicting the topography of the region can be improved further with higher resolution one. Please consider the higher resolution one in the revised version.

Reply: We apologize for the low resolution topography map. We have included a high resolution one in the revised version.

---

## Author Response (AR2)

**Editor's comments reply**

We are thankful to the editor for providing such valuable comments. We accept all the suggestions and corrected our manuscript accordingly.

**Comment 1**_: In the abstract change 'kJkg-1' to 'kJ kg-1'._

**Reply**: We apologize for the typo. We have modified it in the revised version.

**Comment 2:** _line 10: change total air mass with for the whole atmospheric column._

**Reply**: Thank you for pointing this out. The MSE increase is for the lower levels (1000-850 hPa) of the atmosphere. The reason why we used "per total air mass" is because this value is calculated by averaging the lower level MSE both vertically and horizontally for the selected domain. For the whole atmospheric column we first need to integrate the MSE values with respect to pressure or height. We have modified the sentence to
"Results show an average increase in AOD by 36 % along with an average increase in low level moist static energy (1000-850 hPa) by ~ 1500 J $Kg^{-1}$ inside the selected domain for high precipitation events."

**Comment 3:** _line 28 & line 130: 'the consumption of low level MSE', I do not understand this concept. You can reduce the low level MSE but how do you infer that you have a consumption of MSE? Please explain._

**Reply**: We apologize for the confusion. The MSE accounts for the energy available in the atmosphere in the form of heat and moisture (or the summation of internal, latent and potential energies). When there is ample amount of MSE available, convective systems can utilize the MSE (moisture) and the clouds grows into massive precipitating clouds, eventually leading to extreme precipitation events (Fan et al., 2015). The same was meant while addressing "the consumption of low level MSE" in the manuscript. We meant that the energy was used by the convective systems for further development. The presence of absorbing aerosols increases the lower level stability, thereby limiting the energy (heat and moisture) available to the convective systems. Thus less MSE is consumed and is accumulated in the lower levels. This concept is explained schematically in Figure 5 of Fan et al. (2015). Accepting the suggestion, we have modified the lines 26-29 of our manuscript as follows.

"First, the presence of absorbing aerosols in the urban polluted basin increases the lower level stability during the daytime, thereby limiting the moisture in the lower levels. As a result, the moist static energy (MSE) gets accumulated in the low levels. Second, during nighttime, this accumulated MSE is transferred towards the mountains downwind where the now more humid air mass undergoes orographic lifting which results in anomalously high precipitation over the mountains."

**Comment 4:** _line 151: change 'many recent studies' to 'several studies'_

**Reply:** Thank you for pointing this out. We have corrected it in the modified version.

**Comment 5:** _line 180: 'huge amount of aerosols' is very vague and not well formulated. Explicitly say for which concentrations you estimate to have a lot of aerosols (in microgram m-3) or re-formulate_
**Reply:** Since we use aerosol optical depth, an optical parameter to measure the aerosol load, we have modified the sentence as follows:

"The presence of high aerosol load (AOD ~ 0.8) in a profoundly moist environment may also contribute to the excitation of HP events in orographic regions by aerosol indirect effect (Fan et al., 2014, 2017; Xiao et al., 2015)."

**Comment 6:** *lines 208-209: change 'This excess energy, when gets..." to "This excess energy, then gets...".*

**Reply:** Thank you for pointing this out. We have corrected it in the modified manuscript.

[revised manuscript text omitted]